# Demographic amplification is a predictor of invasiveness among plants

Kim Jelbert [1], Danielle Buss [1], Jenni McDonald[1], Stuart Townley [2], Miguel Franco [3], Iain Stott[4], Owen Jones [5], Roberto Salguero-Gómez[6], Yvonne Buckley [7], Tiffany Knight [8,9,10], Matthew Silk [1], Francesca Sargent[1], Simon Rolph [1], Phil Wilson[1] & Dave Hodgson [1*]

Invasive plant species threaten native biodiversity, ecosystems, agriculture, industry and human health worldwide, lending urgency to the search for predictors of plant invasiveness outside native ranges. There is much conflicting evidence about which plant characteristics best predict invasiveness. Here we use a global demographic survey for over 500 plant species to show that populations of invasive plants have better potential to recover from disturbance than non-invasives, even when measured in the native range. Invasives have high stable population growth rates in their invaded ranges, but this metric cannot be predicted based on measurements in the native ranges. Recovery from demographic disturbance is a measure of transient population amplification, linked to high levels of reproduction, and shows phylogenetic signal. Our results demonstrate that transient population dynamics and reproductive capacity can help to predict invasiveness across the plant kingdom, and should guide international policy on trade and movement of plants.

[1] Centre for Ecology and Conservation, University of Exeter, Penryn Campus Penryn Cornwall, UK TR10 9FE. [2] Environment and Sustainability Institute, University of Exeter, Penryn Campus Penryn Cornwall, UK TR10 9FE. [3] School of Biological and Marine Sciences, Plymouth University, Drake Circus, Plymouth PL4 8AA, UK. [4] School of Life Sciences, University of Lincoln, Brayford Pool, Lincoln LN6 7TS, UK. [5] Interdisciplinary Center on Population Dynamics, Department of Biology, University of Southern Denmark, Campusvej 55, 5230 Odense M, Denmark. [6] University of Oxford, Department of Zoology, 11A Mansfield Road, OX1 3SZ Oxford, Oxfordshire, UK. [7] School of Natural Sciences, Zoology, Trinity College Dublin, The University of Dublin, Dublin 2, Ireland. [8] German Centre for Integrative Biodiversity Research (iDiv) Halle-Jena-Leipzig, Deutscher Platz 5e, 04103 Leipzig, Germany. [9] Institute of Biology, Martin Luther University Halle-Wittenberg, Am Kirchtor 1, 06108 Halle, (Saale), Germany. [10] Department of Community Ecology, Helmholtz Centre for Environmental Research- UFZ, Theodor-Lieser-Straße 4, 06120 Halle, (Saale), Germany. *email: d.j.hodgson@exeter.ac.uk

nvasive plant species rank among the most important threats to biodiversity worldwide[1], and are agents of harm to agriculture, industry and human health[2,3]. The importance of invasive species has yielded a large body of scientific endeavour that seeks explanations and predictions for why some species become invasive while others naturalise outside their native range but remain benign[2,4–6]. A variety of ecological approaches have been used to help understand invasiveness among plant species, including functional-trait analyses[6,7] and demographic models[5,8–10]. There is a clear theoretical link between demography and invasiveness, because both are features of population growth and spread, but remarkably few demographic analyses have employed multi-species comparisons (c.f.[5,7,8,10]), in part because of a paucity of accessible data for sufficient numbers of species. The COMPADRE Plant Matrix Database[11], which currently features >8000 stage-structured demographic models representing >700 plant species, means that demographic data is now readily accessible and such comparative analyses are possible.

Any attempt to find predictors of invasiveness must tease apart the constituent features of species' life histories that predict invasiveness, from features of the invaded environment and changes that occur during invasion. Two multi-species, demographic comparisons between invasive and non-invasive species, undertaken to date, have revealed that invasive populations tend to exhibit a stable population growth rate that is higher than both native species in the invaded range[8], and introduced populations of non-invasive congeners with which they co-occur[5]. But these comparative studies, and others[7], suffer two critical limitations. First, they focus on demographic features of invasive species only in their invaded range. This conflates predictors of invasiveness with changes that occur during the invasion process, making it difficult to distinguish between intrinsic invasiveness and changes that are induced by the new environment[12,13]. Second, they include species in their non-invasive categories that are in fact invasive elsewhere in the world. This means that if there is a shared "invasiveness" trait or syndrome among plants, then failure to exclude invasive species from the non-invasive or native categories will weaken or conceal potential predictors of invasiveness.

Here we use a subset of COMPADRE's stage-structured demographic models parameterised with field data from 1201 populations[11] representing 502 plant species, including 175 species that have 'naturalised' outside their native range and 327 species that to our knowledge are restricted to their native range. Of the naturalised species, 113 are non-invasive, and 62 are considered invasive in some part of their naturalised range. We then split populations in each invasiveness category into those studied in the plant's native range, and those studied in their naturalised range. We use these demographic schedules (Supplementary Tables 1 and 2) to present a phylogenetically controlled global, demographic comparison of invasive and non-invasive plant species, seeking predictors of invasiveness based on studies in species' native ranges.

We use two established metrics of stage-structured demographic models, the stable population growth rate ($\lambda$)[14] and demographic inertia ($\rho_\infty$)[15,16], to compare the population dynamics of invasive and non-invasive plants in their native and naturalised ranges. Stable population growth rate measures the population dynamics of populations in undisturbed environments[14]. When disturbed, however, populations can recover quickly or crash, depending on whether the disturbed population structure is biased towards or away from lifestages with high reproductive value. The outcome, and the rate of recovery, is therefore determined not just by the type and intensity of disturbance but also by the life history of the species[16] (Fig. 1). The long-term impact of transient dynamics following demographic

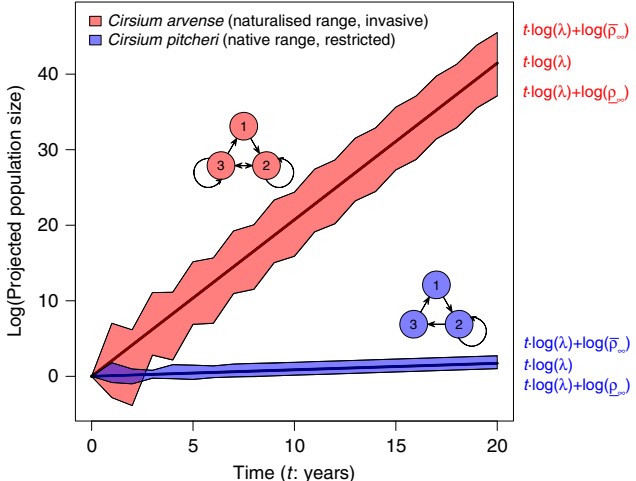

**Fig. 1** Projections of stable and transient population dynamics of two thistle species. *Cirsium pitcheri* (blue) is a non-invasive species whose life cycle in the native range is discretised into three life-stages (1 = seedling, 2 = rosette, 3 = flowering adult)[33] and measured as annual rates of per-capita transition among them. *Cirsium arvense* (red) is invasive (life-stages 1 = seed, 2 = rosette, 3 = flowering adult)[34]. Predicted population dynamics (polygons) are initiated at initial population size of 1 and projected for twenty years. Solid central lines project dynamics of a population initiated at stable stage structure. Polygons capture the envelope of amplification and attenuation achieved by non-stable initial stage structures. Polygon boundaries are functions of time (t), stable rate of increase ($\lambda$), and demographic inertia ($\bar{\rho}_\infty$ and $\rho_\infty$ describing amplification and attenuation, respectively). *C. pitcheri*, the non-invasive species, increases slowly and has a narrow envelope of amplification and attenuation. *C. arvense*, the invasive species, increases rapidly and has a wide envelope of amplification and attenuation. We ask, across plant species, can stable growth rates or demographic inertia, measured in the native range, predict the invasiveness of plants in their naturalised range?.

disturbance is measured by demographic inertia[15], which describes the potential for recovery via long-term population amplification ($\bar{\rho}_\infty$), or failure to recover via population attenuation ($\rho_\infty$), relative to stable growth (Fig. 1 and see "Methods").

We find that the potential to amplify in response to demographic disturbance is a feature of plant life histories that predicts their ability to invade novel environments: demographic inertia is high among invasive plant species, regardless of whether measured in the native or invaded range. We also find that demographic inertia shows phylogenetic signal, and correlates positively with measures of reproductive output. The stable population growth rate is high among invasive plant species, but only when measured in the invaded range.

## Results and discussion

**Demographic predictors of invasiveness.** Invasive plant species exhibit greater potential for demographic amplification ($\bar{\rho}_\infty$) than non-invasive species, in both the native and the naturalised range (Fig. 2a). The potential to recover from demographic disturbance is therefore a species-level trait that differs between invasive species and non-invasive species. In contrast, stable rates of population increase are only high when measured in the naturalised range (Fig. 2b) and so cannot be used as predictors of invasiveness. There are no clear or consistent differences in potential demographic attenuation ($\rho_\infty$) between invasive and non-invasive plant species (Fig. 2c). We suggest that demographic recovery is more relevant to invasiveness than stable growth rates because (1) disturbed environments are known to be more readily

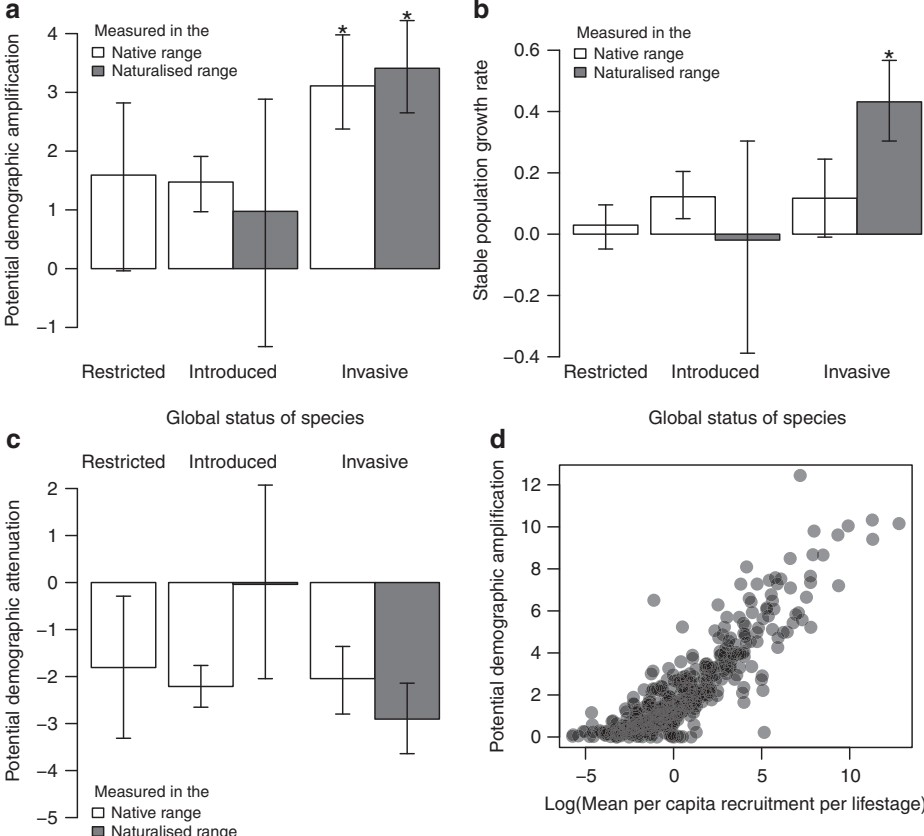

**Fig. 2** Demographic traits of invasive and non-invasive plant species. Bars show mean demographic trait ($+/-95\%$ Credible Intervals) for restricted, introduced and invasive plants, measured in the respective native or naturalised range. Asterisks show invasive categories that are credibly different from all non-invasive categories, based on 95% credible intervals of contrasts between categories not overlapping zero; **a** potential for demographic recovery following disturbance (upper bound on demographic inertia; mean $\log(\bar{\rho}_\infty)$); **b** stable rate of population increase (mean $\log(\lambda)$); and **c** potential for reduced abundance following demographic disturbance (lower bound on demographic inertia; mean $\log(\rho_\infty)$). **d** Relationship between potential demographic recovery (mean $\log(\bar{\rho}_\infty)$) and rates of offspring recruitment (log(mean recruitment per lifestage)). Dark areas are caused by overlapping data. Source data are provided as a Source Data file.

invaded than stable ones[17]; (2) amplified dynamics might promote invasion when populations must grow rapidly to escape Allee effects or demographic stochasticity[18]; and (3) populations with greater potential magnitudes of transient amplification are predicted to grow faster in the short-term and remain larger in the long-term[16], and are therefore more likely to become invasive.

**Correlates of demographic inertia**. The potential for recovery from demographic disturbance correlates strongly and positively with per capita recruitment per life-stage (Fig. 2d). This reinforces the hypothesis that fecundity and seedling survival are useful traits that predict invasiveness[12]. We also find credible signal of phylogenetic patterns in the relationship between demographic amplification and invasiveness (Fig. 3; Supplementary Fig. 1), suggesting that the close relatives of invasive plants share demographic traits that increase their risk of becoming invasive. We find no such signal of phylogenetic patterning in stable population growth rates. We attribute this phylogenetic pattern to the evolutionarily relevant trade-off between seed size and seed number[19]. This phylogenetic patterning is relevant to invasion biology because it suggests that close relatives of invasive plants will be strong candidates for invasiveness if they establish outside their native range. Related species are likely to share invasiveness thanks to their sharing of high potential fecundity and recruitment and therefore demographic amplification. These patterns

suggest that the deliberate export of close relatives of known invasives should be prevented.

**Stable population growth does not predict invasiveness**. Stable rates of population growth are greater in the naturalised range than in the native range, but only among invasive species. This has little value as a predictor of invasiveness, but yields valuable evidence for fundamental changes in the population biology of invasive plants established outside their native range. Explanations for faster stable population growth in the naturalised range include an escape from native natural enemies[20] and competition[21]; genotypic filtering such that only vigorous genotypes establish[22]; an adaptive response to the novel environment of the invaded range[23]; and the possibility that populations in the naturalised range are more likely to have been measured during the rapid establishment phase, than native populations.

**Knowledge gaps and sources of bias**. An important avenue for future research is to strategically collect demographic data for plant species that represent gaps in our knowledge[11]. The first global list of naturalised plant species shows that 13,168 plant species have naturalised outside of their native range[24]. We have demographic data for very few of these species, with a bias towards those that cause environmental harm. There is a critical need to determine if species with proven naturalisation capacity

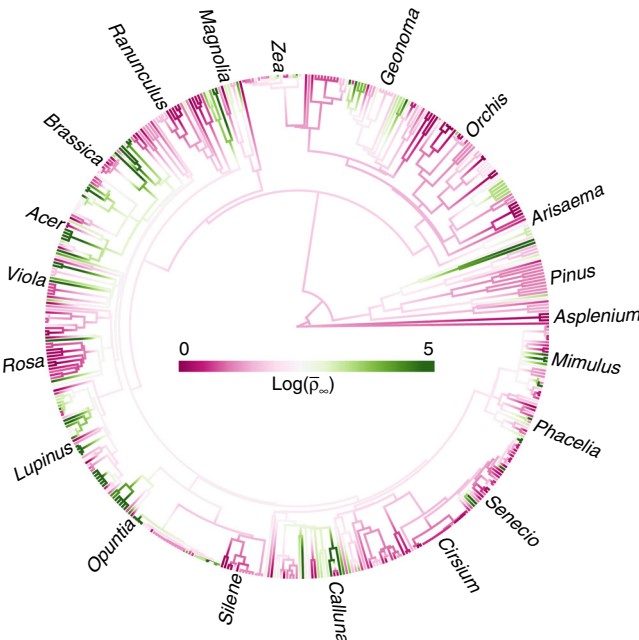

**Fig. 3** Phylogenetic signal in potential to recover from demographic disturbance. Phylogram, showing the magnitude of the upper bound on demographic inertia ($\log(\bar{\rho}_\infty)$) and its distribution across the plant Kingdom. Each tip of the phylogeny represents a species in our dataset. For display purposes, only a subset of 'familiar' genera are labelled. High (green; max = 5) and low (pink; min = 0) values of log(demographic inertia) quantify the potential to recover from demographic disturbance. The clustering of colours across the phylogeny shows that closely related species tend to share similar potential for demographic amplification. Source data are provided as a Source Data file.

are likely to become invasive so that measures can be taken to prevent their introduction and to eradicate existing populations in accordance with target 9 of the IUCN 2020 Strategic Plan for Biodiversity. We suggest targeted research to fill in gaps in the phylogeny, global location, and type of plant species; in particular we need better demographic information on non-invasive alien plants, especially in the naturalised range, for which our current sample size is very small.

Demographic studies in the COMPADRE database are the result of independent investigations conducted for a wide variety of different reasons around the world. We acknowledge the risk that our comparison of invasive and non-invasive species, measured in the native and non-native range, might reflect biases in this global literature on plant demography. First, the matrix models in the COMPADRE database are typically linear and therefore density-independent. Some predictors of invasiveness might be linked to Allee effects, competitive dominance over other species, or other density-dependent processes, and we encourage further research to quantify the role of these population-dynamic nonlinearities as predictors of invasiveness. Second, the empirical study of any plant species' entire life cycle is an intensive piece of research that will often be motivated by the scientific value of the question being posed. In particular, studies of species restricted to their native range are more likely to be motivated by aspects of their ecology that make them interesting per se, rather than their weed or endangered status (Supplementary Table 1). On the other hand, invasive plants studied in their native range might typically be studied there because they may be considered weeds. Such species will contribute "weed" demographies to the COMPADRE database. Only one species in our comparative analysis (*Carduus nutans*) was measured in both the

native and naturalised range, meaning that we are unable to control for species identity (over and above phylogenetic patterning) in our analyses. Despite these biases, scrutiny of the COMPADRE database suggests no associations between demographic metrics (stable growth rate and demographic inertia) and study motivation as cited by the original authors (see "Methods" and Supplementary Tables 1 and 2). Restricted species studied for their conservation value did not differ from restricted species studied for other reasons, and invasive species studied due to their invasiveness or weediness did not differ from invasive species studied for other reasons (Supplementary Table 2). Further, there are no clear biases in the methods used to parameterise demographies of restricted, naturalised and invasive plant species, and we have made every effort to standardise the projection matrices and demographic metrics we use. Nonetheless, we recommend future work to consider species-specific changes in the demography of invasive and non-invasive species between native and naturalised ranges[12].

We have shown that explanations for, and predictions of, invasiveness are found in empirical description of entire life cycles of plant populations growing in native and naturalised ranges. Our comparative database represents a vast amount of work performed by plant ecologists, globally (Supplementary Table 1). Each empirical measure of a population's demography is expensive and time-consuming[11]. An important next step is to simplify the task of predicting invasiveness, for future ecologists, managers and policy-makers, by finding traits or vital rates that are themselves proxies for invasiveness[12,24]. Only some of these traits (e.g. recruitment rates, used here) can be derived from the population projection matrices (PPMs) hosted in the COMPADRE database: much larger comparative datasets should be used to ask questions of functional traits including size, shape and metabolism[25]. We propose that high, stage-structured fecundity is the life history trait that contributes most to the link between demographic amplification and invasiveness. We are not the first to note a link between fecundity and invasiveness (e.g., refs. [5,7]) but its importance as a predictor of demographic amplification, and therefore of invasiveness, is novel and important. We also note that phylogenetic signal in demographic amplification might be explained by phenotypic traits that are clearly patterned by evolutionary history among plants: seed size and fecundity[19]. We recommend deeper exploration of links between seed size, seed production, germination, seedling establishment and invasiveness. We also note that the link between demographic amplification and invasiveness might be caused not just by biological traits that favour invasion, but also by disturbance regimes in invaded and native environments: invasive plants might simply be those that, thanks to being adapted to disturbed environments in the native range, are most suited to disturbed environments in the naturalised range. This means that demographic amplification might not help to predict the identity of invasive species in undisturbed environments.

Our global analysis of plant populations reveals a much needed predictor of invasiveness based on measurements made in the native range. This is important because it will help quarantine authorities to place controls on the export of likely invasive species, thus preventing future invasions. The predictor of invasiveness is yielded not by the classical measure of stable population growth rates, but by the potential for demographic recovery, i.e., amplification in population size following demographic disturbance. Our analyses also link the demography of invasiveness to reproductive traits and phylogenetic relationships among plant species. We recommend that plant species and genera typified by an ability to recover from demographic disturbance, particularly highly fecund species and close relatives of species known to be invasive, should not be exported outside their native range.

## Methods

**Study species and populations, and categorisation**. We extracted all PPMs from the COMPADRE Plant Matrix Database (COMPADRE 3.0.0)[11]. We filtered COMPADRE 3.0.0 by including only matrices that described annual or multi-annual timesteps, and excluding matrices generated by pooling data from multiple sites, and those generated for populations reared in the laboratory or greenhouse. We excluded mean matrices when their constituent, individual matrices were available to use instead, and matrices that were reducible[26]. We also checked all PPMs for the seed-problem[14], in which the seed/propagule stage class is erroneously assumed to last a full year before germination, and where necessary, corrected these. Projection matrices are commonly parameterised as either pre-reproductive (recruitment is measured as fecundity multiplied by rates of germination and seedling survival), or post-reproductive (recruitment of seeds measured as adult survival multiplied by fecundity). Post-reproductive matrices tend to have high values of recruitment, which can affect measures of demographic amplification. We therefore converted all post-reproductive matrices to pre-reproductive matrices using algebraic manipulation of vital rates. Finally, we excluded matrices representing populations that had been manipulated experimentally, for example by treatments associated with burning, herbicide, harvesting, grazing or nutrient supplement. The filtered dataset comprised PPMs representing 1201 spatial populations (many of them replicated through time), representing 502 species of plants (Supplementary Data 1 and 2).

We classified population status for each PPM as either native, invasive, or naturalised but non-invasive, at the location of study, and species status as invasive, naturalised but non-invasive outside of the native range, or restricted to the native range. Population status at the study location was identified from the source literature. Species status outside of the native range was determined by searching invasive species databases (Supplementary Data 1), and by using the following search term in Google: *Latin name* invasive. Species are considered invasive when designated as invasive (also weedy or noxious in the USDA Plant Database) in one or more of the invasive species databases or when designated as invasive by an Academic Institution or Government Agency. Naturalised status was determined by searching the Global Compendium of Weeds (GCW), regional floras and global species distribution databases (Supplementary Data 2). We define naturalised, non-invasive species as those that are naturalised outside of the native range, and restricted species as those that are not known to persist outside their native range. Our refined database includes 32 invasive plant species studied in the naturalised range, 30 invasive plant species studied in the native range, 108 naturalised, non-invasive species studied in the native range, 5 naturalised, non-invasive species studied in the naturalised range and 327 restricted plant species studied in the native range. We simplify the categorisation of plant species to be native or naturalised (i.e. introduced) at the study location; and restricted (never established outside the native range), introduced (established outside the native range but not considered invasive), or invasive (established outside the native range and considered invasive) on a global scale.

**Demographic metrics from PPMs**. The Perron-Frobenius theorem states that the dynamics of a non-negative, irreducible, ergodic projection matrix will, if rates of transition between stages remain constant and growth is not limited, settle from any initial condition to a stable stage structure (relative density of stages in the population) and a stable geometric rate of increase[14]. The stable rate of population increase ($\lambda$) is the dominant eigenvalue of a given population projection matrix and the stable stage structure is the normalised, dominant right eigenvector[14]. If the population is initiated at stable stage structure, then the relationship between abundance ($N$) and time ($t$) is

$$\log(N_t) = \log(N_0) + t\log(\lambda) \tag{1}$$

Demographic inertia ($\rho_\infty$), also known as the Stable Equivalent Ratio[15], measures the long-term impacts of transient population growth or decline caused by disturbance away from stable stage structure[16]. $\rho_\infty$ is the asymptotic ratio of the density of a population disturbed at time zero, to the density of a population initiated at stable stage structure, such that for any initial stage structure:

$$\log(N_t) \to \log(N_0) + t\log(\lambda) + \log(\rho_\infty) \text{ for } t \gg 0 \tag{2}$$

$\rho_\infty$ depends on the population's initial structure, which is usually unavailable in the literature, but it has upper and lower bounds that depend only on the projection matrix itself. We measure both upper and lower bounds on inertia for each matrix model, describing the potential for demographic amplification (more population growth than predicted by $\lambda$) and demographic attenuation (less population growth than predicted by $\lambda$), respectively. In matrix algebra, the upper bound on long-term demographic amplification is

$$\bar{\rho}_\infty = \frac{\mathbf{v}_{\max}\|\mathbf{w}\|_1}{\mathbf{v}^T\mathbf{w}}, \tag{3}$$

where $\mathbf{v}$ is the normalised reproductive value vector (the dominant left eigenvector of the population projection matrix); $\mathbf{v}_{\max}$ is the largest entry in this vector; $\mathbf{w}$ is the stable stage structure (the dominant right eigenvector of the population projection matrix); and $\|\mathbf{w}\|_1$ is the one-norm, i.e. the sum, of the

stable stage structure. Following similar algebra, the lower bound on long-term demographic attenuation is

$$\underline{\rho}_\infty = \frac{\mathbf{v}_{\min}\|\mathbf{w}\|_1}{\mathbf{v}^T\mathbf{w}}, \tag{4}$$

where $\mathbf{v}_{\min}$ is the smallest entry in the reproductive value vector. Useful summaries of measures of transient dynamics are available in the literature[16,27].

**Data handling and analysis**. Our filtered database of projection matrices, representing unmanipulated plant populations, included species that were replicated in space and through time. For each replicate spatial population of each species, we averaged the transition rates through time to create a temporal mean matrix. We calculated demographic metrics (stable rate of increase; upper bound on inertia; lower bound on inertia) per population using these temporal mean matrices. We log-transformed these metrics because they describe geometric processes of population growth or decline, then averaged the metrics across populations to yield means per species per category. We then compared the mean demographic metrics among five categories representing where the species was studied (native versus naturalised range) and their global invasiveness status (restricted, introduced or invasive). This provides a conservative analysis of species-level demographic metrics in relation to invasiveness and study location. Species were non-independent due to phylogenetic history. This hierarchical data structure recommended the use of Monte Carlo Markov Chain general linear mixed-effects modelling, implemented using the MCMCglmm package[28] in R[29]. We used the phylogeny associated with the COMPADRE database, derived from Plantae phylogenies[30,31] by authors TK, RS-G and OJ (Supplementary Software). We set proper uninformative inverse Gamma priors on the error terms associated with residuals, and phylogeny. Log-transformed demographic metrics were modelled with Gaussian error structure. We included parameter expansion terms for the phylogenetic variance, to avoid issues with model convergence. All models were run for 1 million iterations and satisfied standard MCMC diagnostic tests. Code for analyses, and tables of results, are presented in Supplementary Software, alongside the datasets used (Supplementary Data 2). Phylogenetic signal in the residuals was diagnosed by posterior distributions of phylogenetic variance that lay credibly above zero (Supplementary Fig. 1).

Credibility of differences in demographic metrics among invasive categories was determined by testing whether the 95% credible intervals of the contrasts between explanatory variable categories overlapped zero. In our analyses, we contrasted the demographic metrics of invasive species, measured in the naturalised range, against all other categories.

We produced a phylogram that maps the upper bound on demographic inertia through the plant kingdom (Fig. 3), using the contMap function in R library phytools version 0.6-00[32]. This function estimates ancestral states using maximum likelihood based on the rerooting of the tree at each internal node.

**Robustness of results**. The results presented here are for species-level analyses, for which we used mean demographic metrics per species, with phylogenetic control. We chose to present these analyses for their conservatism, their focus on species-level traits relevant to invasiveness, and their simplicity of interpretation. To check robustness of the outcome, we repeated analyses using demographic metrics per population, nested within each species, with the same qualitative results (see Supplementary Fig. 2). We also extended our analyses to the per-population and per-species projection matrices for "experimentally manipulated" populations in COMPADRE, yielding the same outcomes. As a final check of robustness, we performed simple linear mixed-effects modelling of demographic metrics per population per species, and general linear models of metrics per species. These final analyses ignored the phylogenetic patterning of the data, but echoed the results of the MCMCglmm models. Invasiveness is predicted by demographic amplification even in the absence of phylogenetic information. Code and results for these extra analyses are provided in Supplementary Software.

**Checking for bias**. We considered the biases that could be caused by reasons for the study of each plant species by the original authors of the demographic research. Restricted species might be biased towards species studied for their conservation value, and might thus yield fragile demographies, characterised by low rates of population increase, poor survival and low fecundity. Invasive species might be biased towards those studied for their weedy ecology, and might thus yield weedy demographies characterised by high rates of fecundity and rapid population growth. We returned to the original published sources of the projection matrices in COMPADRE, and recorded the "reasons for study" cited by the original authors in their abstracts and introductions. This survey revealed bias among categories in the reasons for study (Supplementary Table 1). We then performed simple Generalised Linear Models to compare the demographic metrics of invasive plants studied for their invasiveness or for other reasons; and GLMs to compare the demographic metrics of restricted plants studied for their endangerment or for other reasons (Supplementary Table 2). We found no association between demography and invasiveness as a reason for study; nor between demography and endangerment as a reason for study.

**Reporting summary**. Further information on research design is available in the Nature Research Reporting Summary linked to this article.

## Data availability

All data used for analyses are provided in online supplementary materials, alongside code for analysis. The COMPADRE database[11] is published online https://www.compadre-db.org/. All data are available from authors on request.

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

## Acknowledgements

We thank the hundreds of ecologists and authors who have contributed demographic data to the COMPADRE database. We thank the COMPADRE database committees and Dave Hosken for advice on the research methodologies and manuscript review. This research was supported by Natural Environment Research Council grant reference NE/L007770/1 and by the University of Exeter's partnership with DEFRA's National Wildlife Management Centre.

## Author contributions

K.J. and D.J.H. compiled the datasets, analysed the data and wrote the manuscript. D.B., F.S., S.R. and P.W. coordinated the data handling. D.B., J.M., S.T. and I.S. contributed to mathematical and data analysis. M.F., M.S., O.J., R.S.G. and Y.B. helped guide the interpretation of results. O.J., R.S.G. and T.K. provided the phylogeny. M.S. provided the Supplementary Material markdown script. All authors edited the manuscript.

## Competing interests

The authors declare no competing interests.
