## [Peer Review File · Nature Communications]

Reviewers' Comments:

Reviewer #1:

Remarks to the Author:

Jelbert et al. reports a study of how invasive plant species' population dynamics compare to that of non-invasive species. The unique aspects of the study hinge on the use of an exceptionally large dataset of population matrices. The underlying research question is what makes some species invasive and others not. The answer to this question has been of wide interest because it should point to interesting biology regarding what allows species to perform well in new areas, and also inform management efforts to minimize the economic and ecological damage from invasive species.

An important aspect of this study, enabled by the large database, is that they are able to analyze invasive species in their native ranges, so they can (to some extent) tease apart inherent traits of the species (in the native range) from conditions that create favourable demography in the introduced range (disturbed habitat, loss of natural enemies, etc). The authors' primary finding is that invasive species have significantly higher demographic amplification even in their native ranges, than species that are not invasive. This is plausible because amplification is related to species' ability to respond demographically after a perturbation, and many invasive species thrive on disturbed habitats. They show that amplification is also positively correlated with per capita recruitment rates, suggesting that this demographic rate may be critical to spread of invasive species. This result is consistent with other studies and meta-analyses also showing that traits such as fitness can help predict invasiveness, but it is novel in its basis in comprehensive demographic analyses. (e.g. the Jelbert et al. 2015 study cited, but also meta-analyses such as Van Kleunen et al. 2010. A meta-analysis of trait differences between invasive and non-invasive plant species. *Ecol. Lett.*),

Overall, this study is a uniquely thorough test of whether invasive species have distinct demographic characteristics that allow them to do well. It is well written and the analyses are excellent and well presented. Some additional explanation of the key demographic inertia metrics would be helpful because they are crucial to the primary result of the paper (particularly amplification). For whatever reason, I have not seen these metrics often in the plant demography literature, and don't have a good feel for them, so further explanation of their calculation and meaning would be helpful.

In sum, I believe the study is an important contribution to this field, and have only a few questions / concerns / points of confusion.

1. Potential biases in dataset

One thing about the analyses that is hard to judge is whether there are any biases in the species or sites for which there is data. For example, I might expect that many of the native species for which there are demographic data are species of conservation concern (e.g. rare, threatened, declining, or otherwise just of conservation interest). I don't have data to support this, but am just thinking about why people tend to collect demographic data on native, "restricted" species. This could introduce a bias toward native "restricted" species having less robust demographic metrics than the invasives.

One idea to explore this potential issue might be to examine whether species' abundances (or range sizes) are predictive of the demographic metrics, even within the native, "restricted" group. Do rare or declining species have less demographic amplification than widespread, abundant natives? These categories of species are of course quite broad, and there may be interesting / important variation within them.

There may be other potential biases in which populations of invasive and naturalised species were studied, but it's hard to judge with a dataset consisting of contributions from many researchers, from

studies with many different purposes. The size of the database is a unique strength; are there ways though to assess any biases. Related, it's a little unclear to me how much overlap in species there is between the 32 'invasive' species studied in the naturalised range and the 30 'invasive' species studied in the native range. Are these the same or different species? If different, does this complicate interpretation of the comparison?

2. I was a bit surprised that the replicate population matrices were just averaged for a species through time to create a "mean matrix". Why not include the spatial variability along with a species effect? There can be significant variability among populations of a species, and it could in fact be interesting to quantify this. The authors in the supplement suggest that such a population-level analysis would be too much to explain in a short paper and that predictions would have to then be made for populations, not for species. I don't fully understand either argument, as it wouldn't have to require a lot more discussion and with the populations nested within species, inference could still be made at the species level it seems to me.

3. Are these all density-independent models? Would inclusion of density-dependence affect the results? I'm considering whether it could help distinguish situations of release in newly disturbed, low-density sites, compared to strong performance in intact communities. I don't know if density can be incorporated at this stage, but it seems like a possible issue to address.

4. Because a conclusion of the study is that fecundity is a plant trait that predicts invasiveness, a broader comparison to previous attempts to identify 'invasive' traits would be helpful (e.g. Van Kleunen et al. 2010 mentioned earlier). Perhaps the authors can explicitly address how their result adds new information to previous efforts. I can imagine some invasion biologists thinking that we already knew that fecundity-related traits were helpful for predicting invasions. I believe the basis in population models here certainly adds strength and mechanistic basis to that claim, but perhaps the authors could address how the conclusion is novel. Related, did the authors consider evaluating more species traits to see if they too are correlated with the demographic metrics? Could add generality (but also could be another paper).

Other questions / suggestions

I gather that there are not annual plant species in these analyses? If not, how might this influence conclusions, given that many of the most invasive species are in fact annuals. Or if there are annuals included, do they show different patterns from the perennial species?

The study reports that it analyzes "1,201 spatial populations (many of them replicated through time)" This confuses me - are there datasets for which there is only a single time point? I don't understand how the matrices were constructed in that case.

A little clearer explanations of notation in equations would be helpful. For example what is λ_m and λ_{mx}

The Bayesian linear mixed-effects models were run for "1 million iterations," which seems like a surprisingly long run. Were there serious convergence or autocorrelation issues, or why run them that long?

What are your "MCMC p-values"? I think I might know, but it would be helpful to spell it out. In the same sentence it is stated that "... determined using MCMC p-values and using overlap of the 95% credible intervals with the means of the poster distributions..." Is this describing two different ways of determining significance, or is that the same way? I could be wrong but always thought calculating the

contrast between two variables (in the model), thereby yielding a posterior of the contrast, was more robust than comparing credible intervals to means afterwards.

Final sentence of the "Robustness of Results" section - I don't follow what you're describing here. Is that "Finally..." analysis an additional analysis, or is that connected to the preceding analyses? Maybe just clearer descriptions here.

In the explanation of why invasive populations may have higher stable growth rates, the authors focus on things related to the biology of the species, but another possibility is the type of sites they are growing in. Many invasive species take advantage of disturbed habitats and may experience less competition / biotic constraints in these sites.

A disclaimer - I have not tried to run the code provided by the authors in the supplement (but applaud them for including it).

Overall, I congratulate the authors on a very interesting study on an important topic, and on the development of an unparalleled database of population models. I hope these comments are constructive.

Signed,
Jeff Diez

Reviewer #3:

Remarks to the Author:

This manuscript addresses the issue of predicting biological invasions. At a time when species are being introduced at an increasing rate, understanding what are the species traits that would make them invasive is critical. Here, the authors used a global demographic data set to explore what those traits are. The analysis shows that species with higher per capita recruitment are more likely to become invasive because they are able to recover rapidly, and grow their populations faster, after disturbance. Since these are traits that can be assessed in the native range, this criterion can be used to identify potential invasive species before the introduction.

Although a useful approach, it may not apply universally. The rates of demographic amplification, although on average higher for invasive species, do not differ among the three groups compared. Thus, it would be difficult to accurately predict, among species with high reproductive outputs, which ones could become invasive. Also, although many invasive events are associated with disturbances, this framework will not help to predict invasions in undisturbed environments.

Where the invasive species used to assess demographic responses in the native and naturalized range the same? If not, could it have been any bias on the species analyzed? i.e., not a random sample of that group. Only 5 species on the naturalized non-invasive category were included, is that a good sample size? Given the 95%CI probably not.

Demographic Predictors of invasiveness among plants

Jelbert et al 2019

Response to Reviewers

Reviewers' comments:

Reviewer #1 (Remarks to the Author):

Jelbert et al. reports a study of how invasive plant species' population dynamics compare to that of non-invasive species. The unique aspects of the study hinge on the use of an exceptionally large dataset of population matrices. The underlying research question is what makes some species invasive and others not. The answer to this question has been of wide interest because it should point to interesting biology regarding what allows species to perform well in new areas, and also inform management efforts to minimize the economic and ecological damage from invasive species.

We're grateful for these positive comments

An important aspect of this study, enabled by the large database, is that they are able to analyze invasive species in their native ranges, so they can (to some extent) tease apart inherent traits of the species (in the native range) from conditions that create favourable demography in the introduced range (disturbed habitat, loss of natural enemies, etc). The authors' primary finding is that invasive species have significantly higher demographic amplification even in their native ranges, than species that are not invasive. This is plausible because amplification is related to species' ability to respond demographically after a perturbation, and many invasive species thrive on disturbed habitats. They show that amplification is also positively correlated with per capita recruitment rates, suggesting that this demographic rate may be critical to spread of invasive species. This result is consistent with other studies and meta-analyses also showing that traits such as fitness can help predict invasiveness, but it is novel in its basis in comprehensive demographic analyses. (e.g. the Jelbert et al. 2015 study cited, but also meta-analyses such as Van Kleunen et al. 2010. A meta-analysis of trait differences between invasive and non-invasive plant species. *Ecol. Lett.*),

Further positive comments, and a good suggestion of Van Kleunen which we have included in our revision.

Overall, this study is a uniquely thorough test of whether invasive species have distinct demographic characteristics that allow them to do well. It is well written and the analyses are excellent and well presented.

Thanks!

Some additional explanation of the key demographic inertia metrics would be helpful because they are crucial to the primary result of the paper (particularly amplification). For whatever reason, I have not seen these metrics often in the plant demography literature, and don't have a good feel for them, so further explanation of their calculation and meaning would be helpful.

We have now provided full mathematical development in Methods, and added citation to a couple of key sources that describe the importance of transient dynamics in plant ecology. We have also added text in the fourth paragraph to justify why demographic amplification might influence

invasiveness.

In sum, I believe the study is an important contribution to this field, and have only a few questions / concerns / points of confusion.

1. Potential biases in dataset

One thing about the analyses that is hard to judge is whether there are any biases in the species or sites for which there is data. For example, I might expect that many of the native species for which there are demographic data are species of conservation concern (e.g. rare, threatened, declining, or otherwise just of conservation interest). I don't have data to support this, but am just thinking about why people tend to collect demographic data on native, "restricted" species. This could introduce a bias toward native "restricted" species having less robust demographic metrics than the invasives.

The risk of bias is hard to avoid in comparative analyses. We have added a whole paragraph to the discussion, which considers the issue of bias. We went back to the primary sources and extracted reasons why each species was studied. In the "restricted" category, many were studied for conservation value but their demographic metrics do not differ from those studied for other reasons (Extended Data Table 4). For the invasives (whether in native or naturalised range), again no difference in demographic metrics between those studied for invasiveness and those studied for other reasons. So, if there are biases in the demographics, they are biases directly associated with the invasive categories, and hence relevant to the analysis. We have included these new data and associated analyses in Extended data, and added a paragraph about tests of bias to the Methods.

One idea to explore this potential issue might be to examine whether species' abundances (or range sizes) are predictive of the demographic metrics, even within the native, "restricted" group. Do rare or declining species have less demographic amplification than widespread, abundant natives? These categories of species are of course quite broad, and there may be interesting / important variation within them.

This is a very interesting idea but we hope this can be seen as a new piece of research that is beyond the scope of this current work. IUCN information on abundance and distribution is available for only ~25% of species in the COMPADRE database, so it would be a major piece of research to bring it together for our full sample of species. It is also an interesting thought exercise to explore what these analyses would prove or disprove. Even if we found that restricted species that were of low abundance or declining distribution had different demographics, it would not really help us to understand the link between demography and invasiveness.

There may be other potential biases in which populations of invasive and naturalised species were studied, but it's hard to judge with a dataset consisting of contributions from many researchers, from studies with many different purposes. The size of the database is a unique strength; are there ways though to assess any biases.

See above. We think we have delivered the best possible check for bias, which is to consider whether demographic metrics differ between species that were studied for applied reasons (e.g. endangerment among restricted species; weediness among invasives) or for basic scientific interest. We find no such differences.

Related, it's a little unclear to me how much overlap in species there is between the 32 'invasive' species studied in the naturalised range and the 30 'invasive' species studied in the native range. Are

these the same or different species? If different, does this complicate interpretation of the comparison?

Only one species was measured in both the native range and the naturalised range. We have added this information to the manuscript (in the discussion about biases). This surprised us when we first did the analysis, and it has prompted further research that tries to increase the number of species' demographics measured in both native and naturalised ranges. This is a major piece of work that can't be included here. While the lack of overlap of species among categories might amplify the risk of bias...it is what it is, the data is as good as it can be, and yet the signal remains.

2. I was a bit surprised that the replicate population matrices were just averaged for a species through time to create a "mean matrix". Why not include the spatial variability along with a species effect? There can be significant variability among populations of a species, and it could in fact be interesting to quantify this. The authors in the supplement suggest that such a population-level analysis would be too much to explain in a short paper and that predictions would have to then be made for populations, not for species. I don't fully understand either argument, as it wouldn't have to require a lot more discussion and with the populations nested within species, inference could still be made at the species level it seems to me.

This is an interesting point. We have performed and presented analyses with populations averaged out as pseudoreplicates of each species. This deals with the nonindependence of populations within species. We have also performed, and now present in Extended Data, an analysis of populations nested within species. The same patterns emerge at the population level. The full hierarchical analysis suggested by the reviewer, with full phylogenetic control and inference at the species level, would be a complex analysis requiring us to introduce polytomies at the tips of the phylogeny then extract marginal residuals for species and for populations. While we agree that variation among populations (and through time) is interesting, we also think that such an analysis would push the paper beyond its remit. Other researchers have published, or are working on, analyses of demographic variation in space and time.

In particular we note conclusions made by Coutts, S. R., Salguero-Gómez, R., Csergő, A. M., & Buckley, Y. M. (2016). Extrapolating demography with climate, proximity and phylogeny: approach with caution. *Ecology letters*, 19(12), 1429-1438. They showed that replicate population of species in COMPADRE tend to be very close in proximity, and tend to share similar demographics. Our "per species" analysis is a conservative approach that still yields impressive signal for the predictiveness of invasiveness related to demographic amplification. We hope the reviewer agrees that the "averaging-out" analysis is better fit to the storyline, and provides results that are robust to the inclusion of populations.

We have clarified the conservatism of our approach in the Methods section, and clarified our paragraph about "robustness checks". However we model this system, we get strong signal of demographic amplification as a predictor of invasiveness.

3. Are these all density-independent models? Would inclusion of density-dependence affect the results? I'm considering whether it could help distinguish situations of release in newly disturbed, low-density sites, compared to strong performance in intact communities. I don't know if density can be incorporated at this stage, but it seems like a possible issue to address.

Population projection models are nearly always linear and therefore density-independent. Authors very rarely report densities when they report projection matrices (to our immense frustration). To

some extent, variation in density is dealt with by averaging projection matrices through time, although there is considerable variation in the length of each time series. We have added DI as a caveat to our new discussion paragraph.

4. Because a conclusion of the study is that fecundity is a plant trait that predicts invasiveness, a broader comparison to previous attempts to identify 'invasive' traits would be helpful (e.g. Van Kleunen et al. 2010 mentioned earlier). Perhaps the authors can explicitly address how their result adds new information to previous efforts. I can imagine some invasion biologists thinking that we already knew that fecundity-related traits were helpful for predicting invasions. I believe the basis in population models here certainly adds strength and mechanistic basis to that claim, but perhaps the authors could address how the conclusion is novel.

Good point. We provide information in the introduction regarding other efforts to find traits that predict invasiveness. We have now added information to the discussion, pointing out that fecundity is perhaps not novel as a proposed explanation of invasiveness. This is, however, the first time that it has been revealed as a predictor based on measurements in the native range.

Related, did the authors consider evaluating more species traits to see if they too are correlated with the demographic metrics? Could add generality (but also could be another paper).

Other "traits" are hard to source from COMPADRE data, since the demographics tend to include only stage-structured rates of survival, growth and recruitment, and indeed the entries in the matrices are usually compound functions of these vital rates. We chose to present "recruitment" even though it is itself a compound trait of seed production, germination and early life survival. Our thesis is that the linking of phenotypic traits to invasiveness should use larger databases such as TRY.

Other questions / suggestions

I gather that there are not annual plant species in these analyses? If not, how might this influence conclusions, given that many of the most invasive species are in fact annuals. Or if there are annuals included, do they show different patterns from the perennial species?

Annuals are in fact included in COMPADRE if they have a seed bank whose survival and germination rates have been measured. It's true though that the majority of COMPADRE species are perennials. We're also not sure that most invasives are annuals. Analyses of demographic metrics in relation to plant growth form, perennation and other traits have been performed and published (e.g. Salguero-Gómez, R. (2017). Applications of the fast-slow continuum and reproductive strategy framework of plant life histories. *New Phytologist*, 213(4), 1618-1624.) but we prefer to reserve links between annuality and demography and invasiveness for future work.

The study reports that it analyzes "1,201 spatial populations (many of them replicated through time)" This confuses me - are there datasets for which there is only a single time point? I don't understand how the matrices were constructed in that case.

Apologies...this is lack of clarity in our writing. A projection matrix can only be parameterised with at least two time points. We have revised the text to read "many of them replicated over multiple timesteps"

A little clearer explanations of notation in equations would be helpful. For example what is λ_m and λ_{mx}

We have revised all formulae for consistent use of the metrics, and have added algebraic content to the Methods section to help clarify. In particular, λ_m and λ_{mx} were scruffy writing of the “lambda” we use everywhere else in the manuscript.

The Bayesian linear mixed-effects models were run for “1 million iterations,” which seems like a surprisingly long run. Were there serious convergence or autocorrelation issues, or why run them that long?

The long chains were used because the large dataset and large phylogeny risked the inflation of residual error and parameter estimates. We performed all standard model checks to confirm convergence and lack of autocorrelation. Also, we like to be really thorough with our MCMC analyses.

What are your “MCMC p-values”? I think I might know, but it would be helpful to spell it out. In the same sentence it is stated that “... determined using MCMC p-values and using overlap of the 95% credible intervals with the means of the poster distributions...” Is this describing two different ways of determining significance, or is that the same way? I could be wrong but always thought calculating the contrast between two variables (in the model), thereby yielding a posterior of the contrast, was more robust than comparing credible intervals to means afterwards.

Great point. We were lazy with our description of the tests used for “significance”. We have revised the text in Methods to state “Credibility of differences in demographic metrics among invasiveness categories was determined by testing whether the 95% credible intervals of the contrasts between explanatory variable categories overlapped zero”

Final sentence of the “Robustness of Results” section - I don’t follow what you’re describing here. Is that “Finally...” analysis an additional analysis, or is that connected to the preceding analyses? Maybe just clearer descriptions here.

We have improved this paragraph to clarify that the extra analyses were included to check that the outcome was not a feature of the statistical engine. Even without phylogenetic information, the signal remains. We have also used this paragraph to introduce and explain the per-species analysis provided in Extended Data.

In the explanation of why invasive populations may have higher stable growth rates, the authors focus on things related to the biology of the species, but another possibility is the type of sites they are growing in. Many invasive species take advantage of disturbed habitats and may experience less competition / biotic constraints in these sites.

We have added text to this effect in last few paragraphs. In fact there is considerable debate in the plant ecology literature regarding whether invasive species are invasive mainly in disturbed habitats. We are constrained by number of references for this journal, but hope that our discussion text helps to include this possibility.

A disclaimer - I have not tried to run the code provided by the authors in the supplement (but applaud them for including it).

Overall, I congratulate the authors on a very interesting study on an important topic, and on the development of an unparalleled database of population models. I hope these comments are constructive.

We are extremely grateful for the perceptive and constructive review.

Signed,
Jeff Diez

Reviewer #3 (Remarks to the Author):

This manuscript addresses the issue of predicting biological invasions. At a time when species are being introduced at an increasing rate, understanding what are the species traits that would make them invasive is critical. Here, the authors used a global demographic data set to explore what those traits are. The analysis shows that species with higher per capita recruitment are more likely to become invasive because they are able to recover rapidly, and grow their populations faster, after disturbance. Since these are traits that can be assessed in the native range, this criterion can be used to identified potential invasive species before the introduction.

Although a useful approach, it may not apply universally.

While true, we remain very excited that we have found such strong signal in our metrics of transient dynamics. Of course no statistical prediction of invasiveness will ever be perfect.

The rates of demographic amplification, although in average higher for invasive species, do not differ among the three groups compared.

We are not sure what is meant by this critique. 95% Credible Intervals clearly show that demographic amplification is credibly higher among invasive species measured in either the native or naturalised range (Figure 2A). We provide 95%CRIs and contrasts to support this.

Thus, it would be difficult to accurately predict, among species with high reproductive outputs, which ones could become invasive.

This is true, but the signal is strong and the precautionary principle implies that all species with high rates of recruitment, yielding high potential to recover from disturbance, should be considered potentially invasive outside the native range. We do not claim accurate predictions...instead we claim credible predictive signal.

Also, although many invasive events are associated with disturbances, these framework will not help to predict invasions in undisturbed environments.

An interesting point linked to the bias arguments made by Reviewer #1. We have included a statement to this effect in the discussion.

Where the invasive species used to assess demographic responses in the native and naturalized range the same?

See response to Reviewer #1. Only one species was measured in both the native and naturalised ranges.

If not, could it have been any bias on the species analyzed? i.e., not a random sample of that group.

See our response to the risk of bias, above

Only 5 species on the naturalized non-invasive category were included, is that a good sample size? Given the 95%CI probably not.

We agree that this is a small sample size, however this category is not essential to the conclusions of the paper. Much more interesting is the comparison of invasives to “restricted” and “non-invasive but measured in the native range”. In the discussion we recommend strategic demography-measurement of plants, particularly in this category of naturalised, non-invasive plants.

Reviewers' Comments:

Reviewer #1:

Remarks to the Author:

I thought the authors' responses to the first round of comments were fair and well-reasoned, and that the changes made to the manuscript were good. In particular the added discussion of potential biases is an important addition and helps put these results in context. Overall, I'm satisfied with the responses and edits, and I still believe the study is a highly novel, interesting contribution to the field. The study also points to a number of potentially fruitful avenues for future research.

Jeff

Reviewer #3:

Remarks to the Author:

In this study, by analyzing a very extensive data set, the authors provide evidence for a mechanism of invasion that can be inferred from performance in the native range. The potential for recovery from demographic disturbances is a better predictor of invasive potential than stable rates of population increase. Furthermore, authors also document a phylogenetic pattern of this trait, which may also help to identify potential invaders.

There is no doubt that understanding the mechanisms underlying successful biological invasions is a pressing research topic in ecology. Most of the controversies around this topic arise because the mechanisms that may have driven one particular invasion event may not apply in a different context. Thus, the field is restricted to broad generalizations about biological invasions, but still constrained about predicting individual events. This study will lessen some of those constraints by helping to assess the invasive potential of plant species on the basis of their performance in their native ranges.

Make title more informative and related to results.

L27-39 It is difficult to figure out the key novelty of this analysis, from that sentence. May be use "Our analysis demonstrated that high reproductive capacity after demographic disturbance is a good, phylogenetically related, predictor of invasiveness across the ..."

L30 Not sure there is little consensus. I believe there is consensus, the issue is that there is no one trait or factor driving the invasion, but many, and these are context dependent. One well-known trait is high population growth rate (through high reproductive output); what is not that clear is when, i.e., the context, high population growth rates will lead to a successful invasion event or not.

Fig. 2 It is not entirely clear what the contrast (*) is, Invasive in the introduced range vs introduced in the introduced range? Or/and, invasive in the native range vs introduced in the native range? L339-341 don't specify either.

Demographic amplification is a predictor of invasiveness among plants

Jelbert et al 2019

Response to Reviewers

1. Final Revisions

REVIEWERS' COMMENTS:

Reviewer #1 (Remarks to the Author):

I thought the authors' responses to the first round of comments were fair and well-reasoned, and that the changes made to the manuscript were good. In particular the added discussion of potential biases is an important addition and helps put these results in context. Overall, I'm satisfied with the responses and edits, and I still believe the study is a highly novel, interesting contribution to the field. The study also points to a number of potentially fruitful avenues for future research.
Jeff

We're grateful.

Reviewer #3 (Remarks to the Author):

In this study, by analyzing a very extensive data set, the authors provide evidence for a mechanism of invasion that can be inferred from performance in the native range. The potential for recovery from demographic disturbances is a better predictor of invasive potential than stable rates of population increase. Furthermore, authors also document a phylogenetic pattern of this trait, which may also help to identify potential invaders.

There is no doubt that understanding the mechanisms underlying successful biological invasions is a pressing research topic in ecology. Most of the controversies around this topic arise because the mechanisms that may have driven one particular invasion event may not apply in a different context. Thus, the field is restricted to broad generalizations about biological invasions, but still constrained about predicting individual events. This study will lessen some of those constraints by helping to assess the invasive potential of plant species on the basis of their performance in their native ranges.

Make title more informative and related to results.

We are happy to change the title to "Demographic amplification is a predictor of invasiveness among plants"

L27-39 It is difficult to figure out the key novelty of this analysis, from that sentence. May be use "Our analysis demonstrated that high reproductive capacity after demographic disturbance is a good, phylogenetically related, predictor of invasiveness across the ..."

We have taken advice from the editor on the novelty statements in our abstract and introduction.

We have also unpacked the novelty and findings in the last paragraph of the introduction, to read: We find that the potential to amplify in response to demographic disturbance is a feature of plant life histories that predicts their ability to invade novel environments: demographic inertia is high among invasive plant species, regardless of whether measured in the native or invaded range. We also find that demographic inertia shows phylogenetic signal, and correlates positively with measures of reproductive output. The stable population growth rate is high among invasive plant species, but only when measured in the invaded range.

L30 Not sure there is little consensus. I believe there is consensus, the issue is that there is no one trait or factor driving the invasion, but many, and these are context dependent. One well-known trait is high population growth rate (through high reproductive output); what is not that clear is when, i.e., the context, high population growth rates will lead to a successful invasion event or not.

We have removed reference to “lack of consensus”, even though we believe it is valid to seek a trait that consistently predicts invasiveness.

Fig. 2 It is not entirely clear what the contrast (*) is, Invasive in the introduced range vs introduced in the introduced range? Or/and, invasive in the native range vs introduced in the native range? L339-341 don't specify either.

The caption for Figure 2 now clarifies that the contrasts are between invasive categories and all other categories. This is also clarified in the Methods text.

2. Previous revisions:

Reviewer #1 (Remarks to the Author):

Jelbert et al. reports a study of how invasive plant species' population dynamics compare to that of non-invasive species. The unique aspects of the study hinge on the use of an exceptionally large dataset of population matrices. The underlying research question is what makes some species invasive and others not. The answer to this question has been of wide interest because it should point to interesting biology regarding what allows species to perform well in new areas, and also inform management efforts to minimize the economic and ecological damage from invasive species.

We're grateful for these positive comments

An important aspect of this study, enabled by the large database, is that they are able to analyze invasive species in their native ranges, so they can (to some extent) tease apart inherent traits of the species (in the native range) from conditions that create favourable demography in the introduced range (disturbed habitat, loss of natural enemies, etc). The authors' primary finding is that invasive species have significantly higher demographic amplification even in their native ranges, than species that are not invasive. This is plausible because amplification is related to species' ability to respond demographically after a perturbation, and many invasive species thrive on disturbed habitats. They show that amplification is also positively correlated with per capita recruitment rates, suggesting that this demographic rate may be critical to spread of invasive species. This result is consistent with

other studies and meta-analyses also showing that traits such as fitness can help predict invasiveness, but it is novel in its basis in comprehensive demographic analyses. (e.g. the Jelbert et al. 2015 study cited, but also meta-analyses such as Van Kleunen et al. 2010. A meta-analysis of trait differences between invasive and non-invasive plant species. *Ecol. Lett.*),

Further positive comments, and a good suggestion of Van Kleunen which we have included in our revision.

Overall, this study is a uniquely thorough test of whether invasive species have distinct demographic characteristics that allow them to do well. It is well written and the analyses are excellent and well presented.

Thanks!

Some additional explanation of the key demographic inertia metrics would be helpful because they are crucial to the primary result of the paper (particularly amplification). For whatever reason, I have not seen these metrics often in the plant demography literature, and don't have a good feel for them, so further explanation of their calculation and meaning would be helpful.

We have now provided full mathematical development in Methods, and added citation to a couple of key sources that describe the importance of transient dynamics in plant ecology. We have also added text in the fourth paragraph to justify why demographic amplification might influence invasiveness.

In sum, I believe the study is an important contribution to this field, and have only a few questions / concerns / points of confusion.

1. Potential biases in dataset

One thing about the analyses that is hard to judge is whether there are any biases in the species or sites for which there is data. For example, I might expect that many of the native species for which there are demographic data are species of conservation concern (e.g. rare, threatened, declining, or otherwise just of conservation interest). I don't have data to support this, but am just thinking about why people tend to collect demographic data on native, "restricted" species. This could introduce a bias toward native "restricted" species having less robust demographic metrics than the invasives.

The risk of bias is hard to avoid in comparative analyses. We have added a whole paragraph to the discussion, which considers the issue of bias. We went back to the primary sources and extracted reasons why each species was studied. In the "restricted" category, many were studied for conservation value but their demographic metrics do not differ from those studied for other reasons (Extended Data Table 4). For the invasives (whether in native or naturalised range), again no difference in demographic metrics between those studied for invasiveness and those studied for other reasons. So, if there are biases in the demographics, they are biases directly associated with the invasive categories, and hence relevant to the analysis. We have included these new data and associated analyses in Extended data, and added a paragraph about tests of bias to the Methods.

One idea to explore this potential issue might be to examine whether species' abundances (or range sizes) are predictive of the demographic metrics, even within the native, "restricted" group. Do rare or declining species have less demographic amplification than widespread, abundant natives? These categories of species are of course quite broad, and there may be interesting / important variation within them.

This is a very interesting idea but we hope this can be seen as a new piece of research that is beyond the scope of this current work. IUCN information on abundance and distribution is available for only ~25% of species in the COMPADRE database, so it would be a major piece of research to bring it together for our full sample of species. It is also an interesting thought exercise to explore what these analyses would prove or disprove. Even if we found that restricted species that were of low abundance or declining distribution had different demographics, it would not really help us to understand the link between demography and invasiveness.

There may be other potential biases in which populations of invasive and naturalised species were studied, but it's hard to judge with a dataset consisting of contributions from many researchers, from studies with many different purposes. The size of the database is a unique strength; are there ways though to assess any biases.

See above. We think we have delivered the best possible check for bias, which is to consider whether demographic metrics differ between species that were studied for applied reasons (e.g. endangerment among restricted species; weediness among invasives) or for basic scientific interest. We find no such differences.

Related, it's a little unclear to me how much overlap in species there is between the 32 'invasive' species studied in the naturalised range and the 30 'invasive' species studied in the native range. Are these the same or different species? If different, does this complicate interpretation of the comparison?

Only one species was measured in both the native range and the naturalised range. We have added this information to the manuscript (in the discussion about biases). This surprised us when we first did the analysis, and it has prompted further research that tries to increase the number of species' demographics measured in both native and naturalised ranges. This is a major piece of work that can't be included here. While the lack of overlap of species among categories might amplify the risk of bias...it is what it is, the data is as good as it can be, and yet the signal remains.

2. I was a bit surprised that the replicate population matrices were just averaged for a species through time to create a "mean matrix". Why not include the spatial variability along with a species effect? There can be significant variability among populations of a species, and it could in fact be interesting to quantify this. The authors in the supplement suggest that such a population-level analysis would be too much to explain in a short paper and that predictions would have to then be made for populations, not for species. I don't fully understand either argument, as it wouldn't have to require a lot more discussion and with the populations nested within species, inference could still be made at the species level it seems to me.

This is an interesting point. We have performed and presented analyses with populations averaged out as pseudoreplicates of each species. This deals with the nonindependence of populations within species. We have also performed, and now present in Extended Data, an analysis of populations nested within species. The same patterns emerge at the population level. The full hierarchical analysis suggested by the reviewer, with full phylogenetic control and inference at the species level, would be a complex analysis requiring us to introduce polytomies at the tips of the phylogeny then extract marginal residuals for species and for populations. While we agree that variation among populations (and through time) is interesting, we also think that such an analysis would push the paper beyond its remit. Other researchers have published, or are working on, analyses of demographic variation in space and time.

In particular we note conclusions made by Coutts, S. R., Salguero-Gómez, R., Csergő, A. M., & Buckley, Y. M. (2016). Extrapolating demography with climate, proximity and phylogeny: approach with caution. *Ecology letters*, 19(12), 1429-1438. They showed that replicate population of species in COMPADRE tend to be very close in proximity, and tend to share similar demographies. Our “per species” analysis is a conservative approach that still yields impressive signal for the predictiveness of invasiveness related to demographic amplification. We hope the reviewer agrees that the “averaging-out” analysis is better fit to the storyline, and provides results that are robust to the inclusion of populations.

We have clarified the conservatism of our approach in the Methods section, and clarified our paragraph about “robustness checks”. However we model this system, we get strong signal of demographic amplification as a predictor of invasiveness.

3. Are these all density-independent models? Would inclusion of density-dependence affect the results? I’m considering whether it could help distinguish situations of release in newly disturbed, low-density sites, compared to strong performance in intact communities. I don’t know if density can be incorporated at this stage, but it seems like a possible issue to address.

Population projection models are nearly always linear and therefore density-independent. Authors very rarely report densities when they report projection matrices (to our immense frustration). To some extent, variation in density is dealt with by averaging projection matrices through time, although there is considerable variation in the length of each time series. We have added DI as a caveat to our new discussion paragraph.

4. Because a conclusion of the study is that fecundity is a plant trait that predicts invasiveness, a broader comparison to previous attempts to identify ‘invasive’ traits would be helpful (e.g. Van Kleunen et al. 2010 mentioned earlier). Perhaps the authors can explicitly address how their result adds new information to previous efforts. I can imagine some invasion biologists thinking that we already knew that fecundity-related traits were helpful for predicting invasions. I believe the basis in population models here certainly adds strength and mechanistic basis to that claim, but perhaps the authors could address how the conclusion is novel.

Good point. We provide information in the introduction regarding other efforts to find traits that predict invasiveness. We have now added information to the discussion, pointing out that fecundity is perhaps not novel as a proposed explanation of invasiveness. This is, however, the first time that it has been revealed as a predictor based on measurements in the native range.

Related, did the authors consider evaluating more species traits to see if they too are correlated with the demographic metrics? Could add generality (but also could be another paper).

Other “traits” are hard to source from COMPADRE data, since the demographies tend to include only stage-structured rates of survival, growth and recruitment, and indeed the entries in the matrices are usually compound functions of these vital rates. We chose to present “recruitment” even though it is itself a compound trait of seed production, germination and early life survival. Our thesis is that the linking of phenotypic traits to invasiveness should use larger databases such as TRY.

Other questions / suggestions

I gather that there are not annual plant species in these analyses? If not, how might this influence

conclusions, given that many of the most invasive species are in fact annuals. Or if there are annuals included, do they show different patterns from the perennial species?

Annuals are in fact included in COMPADRE if they have a seed bank whose survival and germination rates have been measured. It's true though that the majority of COMPADRE species are perennials. We're also not sure that most invasives are annuals. Analyses of demographic metrics in relation to plant growth form, perennation and other traits have been performed and published (e.g. Salguero-Gómez, R. (2017). Applications of the fast-slow continuum and reproductive strategy framework of plant life histories. *New Phytologist*, 213(4), 1618-1624.) but we prefer to reserve links between annuality and demography and invasiveness for future work.

The study reports that it analyzes "1,201 spatial populations (many of them replicated through time)" This confuses me - are there datasets for which there is only a single time point? I don't understand how the matrices were constructed in that case.

Apologies...this is lack of clarity in our writing. A projection matrix can only be parameterised with at least two time points. We have revised the text to read "many of them replicated over multiple timesteps"

A little clearer explanations of notation in equations would be helpful. For example what is λ_m and λ_{mx}

We have revised all formulae for consistent use of the metrics, and have added algebraic content to the Methods section to help clarify. In particular, λ_m and λ_{mx} were scruffy writing of the "lambda" we use everywhere else in the manuscript.

The Bayesian linear mixed-effects models were run for "1 million iterations," which seems like a surprisingly long run. Were there serious convergence or autocorrelation issues, or why run them that long?

The long chains were used because the large dataset and large phylogeny risked the inflation of residual error and parameter estimates. We performed all standard model checks to confirm convergence and lack of autocorrelation. Also, we like to be really thorough with our MCMC analyses.

What are your "MCMC p-values"? I think I might know, but it would be helpful to spell it out. In the same sentence it is stated that "... determined using MCMC p-values and using overlap of the 95% credible intervals with the means of the poster distributions..." Is this describing two different ways of determining significance, or is that the same way? I could be wrong but always thought calculating the contrast between two variables (in the model), thereby yielding a posterior of the contrast, was more robust than comparing credible intervals to means afterwards.

Great point. We were lazy with our description of the tests used for "significance". We have revised the text in Methods to state "Credibility of differences in demographic metrics among invasiveness categories was determined by testing whether the 95% credible intervals of the contrasts between explanatory variable categories overlapped zero"

Final sentence of the "Robustness of Results" section - I don't follow what you're describing here. Is that "Finally..." analysis an additional analysis, or is that connected to the preceding analyses?

Maybe just clearer descriptions here.

We have improved this paragraph to clarify that the extra analyses were included to check that the outcome was not a feature of the statistical engine. Even without phylogenetic information, the signal remains. We have also used this paragraph to introduce and explain the per-species analysis provided in Extended Data.

In the explanation of why invasive populations may have higher stable growth rates, the authors focus on things related to the biology of the species, but another possibility is the type of sites they are growing in. Many invasive species take advantage of disturbed habitats and may experience less competition / biotic constraints in these sites.

We have added text to this effect in last few paragraphs. In fact there is considerable debate in the plant ecology literature regarding whether invasive species are invasive mainly in disturbed habitats. We are constrained by number of references for this journal, but hope that our discussion text helps to include this possibility.

A disclaimer - I have not tried to run the code provided by the authors in the supplement (but applaud them for including it).

Overall, I congratulate the authors on a very interesting study on an important topic, and on the development of an unparalleled database of population models. I hope these comments are constructive.

We are extremely grateful for the perceptive and constructive review.

Signed,
Jeff Diez

Reviewer #3 (Remarks to the Author):

This manuscript addresses the issue of predicting biological invasions. At a time when species are being introduced at an increasing rate, understanding what are the species traits that would make them invasive is critical. Here, the authors used a global demographic data set to explore what those traits are. The analysis shows that species with higher per capita recruitment are more likely to become invasive because they are able to recover rapidly, and grow their populations faster, after disturbance. Since these are traits that can be assessed in the native range, this criterion can be used to identified potential invasive species before the introduction.

Although a useful approach, it may not apply universally.

While true, we remain very excited that we have found such strong signal in our metrics of transient dynamics. Of course no statistical prediction of invasiveness will ever be perfect.

The rates of demographic amplification, although in average higher for invasive species, do not differ among the three groups compared.

We are not sure what is meant by this critique. 95% Credible Intervals clearly show that demographic amplification is credibly higher among invasive species measured in either the native or naturalised range (Figure 2A). We provide 95%CRIs and contrasts to support this.

Thus, it would be difficult to accurately predict, among species with high reproductive outputs, which ones could become invasive.

This is true, but the signal is strong and the precautionary principle implies that all species with high rates of recruitment, yielding high potential to recover from disturbance, should be considered potentially invasive outside the native range. We do not claim accurate predictions...instead we claim credible predictive signal.

Also, although many invasive events are associated with disturbances, these framework will not help to predict invasions in undisturbed environments.

An interesting point linked to the bias arguments made by Reviewer #1. We have included a statement to this effect in the discussion.

Where the invasive species used to assess demographic responses in the native and naturalized range the same?

See response to Reviewer #1. Only one species was measured in both the native and naturalised ranges.

If not, could it have been any bias on the species analyzed? i.e., not a random sample of that group.

See our response to the risk of bias, above

Only 5 species on the naturalized non-invasive category where included, is that a good sample size? Given the 95%CI probably not.

We agree that this is a small sample size, however this category is not essential to the conclusions of the paper. Much more interesting is the comparison of invasives to “restricted” and “non-invasive but measured in the native range”. In the discussion we recommend strategic demography-measurement of plants, particularly in this category of naturalised, non-invasive plants.